# Sparse Training: Do All Tokens Matter for Long Sequence Generalization?

## Abstract

Large language models (LLMs) have demonstrated remarkable progress in generating high-quality natural language through extensive pre-training over Transformer architectures. However, the quadratic complexity of transformers in sequence computation greatly limits their capability to efficiently model long sequences. In this paper, we introduce Sparse Training, a simple training technique to optimize the complexity of Transformer models in long-sequence generalization. Specifically, in Sparse Training, the input sequences of the Transformer network are segmented into two distinct components: the *memory* part and the *target* part. The target part adheres to the standard next-token prediction for modeling continuous sequences, while the memory part, sampled from longer sequences, serves as the conditional context for the prediction of the target part. To build the memory part, we apply a sparse sampling policy that decays with the distance from the target part, to obtain tokens and preserve their positions. Without any architectural modifications, our method can extend existing Transformer-based LLMs to capture long-range dependencies within a fixed window size during the training. Experimental results on multiple datasets also demonstrate the effectiveness and efficiency of Sparse Training to mitigate the complexity of the Transformer network in building long-sequence dependency.

## 1 Introduction

With the aid of large-scale pre-training techniques (Kaplan et al., 2020; Ouyang et al., 2022) on the Transformer models (Vaswani et al., 2017), large language models (LLMs) (OpenAI, 2023; Touvron et al., 2023a;b; Team, 2024a; Jiang et al., 2023; Team & Google, 2023; Team, 2024b) have recently achieved incredible progress in solving massive natural language processing (NLP) tasks (e.g., generation, reasoning, translation, etc). Despite these remarkable advancements, the inherent issue of quadratic complexity in the Transformer networks severely limits their capability to extend long-sequence modeling, drawing enormous attention from both the industry and academia to address this critical issue.

Generally, many efforts have been devoted to generalizing the context windows of LLMs beyond their pre-training settings. Among these works, some researchers attempted to develop sparse architectures (Child et al., 2019; Beltagy et al., 2020; Zaheer et al., 2020; Choromanski et al., 2021; Tay et al., 2023; Han et al., 2024; Xiao et al., 2024) to reduce the quadratic complexity of Transformer network during the training phase. However, these architectures involve sparse patterns and limit their scalability to fall behind the original ones. Therefore, further works continue to explore how to extend existing LLMs to support long-sequence dependency. To this end, some papers (e.g., RoPE (Su et al., 2024), ALiBi (Press et al., 2022), LEX-Transformer (Sun et al., 2023b)) point out that good positional information plays an important role in enabling length extrapolation. On the basis of these, some papers (e.g., PI (Chen et al., 2023), Yarn (Peng et al., 2024)) extend positional information to enlarge context windows via interpolation. Although these works offer a solid initialization for modeling positional information in long sequences, they still experience performance deterioration without any fine-tuning. How to devise an efficient training method to extend the context window of existing LLMs still remains an ongoing challenge.

In this paper, inspired by previous experiences (Child et al., 2019; Beltagy et al., 2020; Zaheer et al., 2020; Choromanski et al., 2021; Tay et al., 2023; Han et al., 2024; Xiao et al., 2024), we observe and analyze the phenomenon of attention sparsity, particularly in long-sequence modeling, and further attribute it as "Pareto Principle of Transformers". That is, only a small subset of tokens dominates

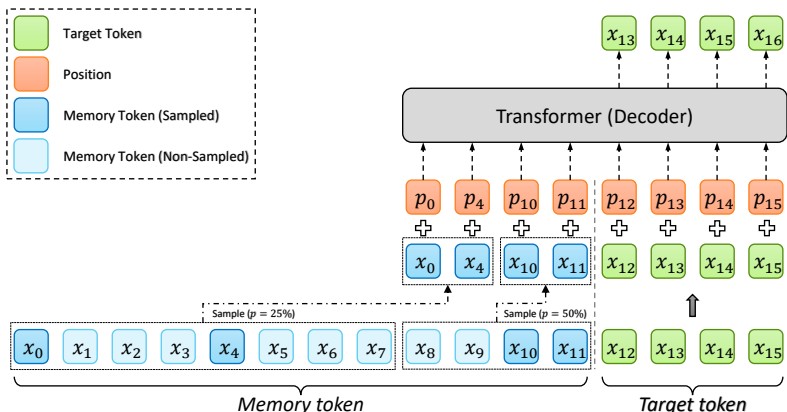

Figure 1: The example of SPARSE TRAINING. Assume the window size of this language model is 8. We expect to sample 8 tokens from a document with 16 tokens to simulate training. Here, we divide the input document as the memory part ($x_{0-11}$) and the target part ($x_{12-15}$). Then, we sample ($x_0$, $x_4$) from ($x_{0-7}$) with a probability of 25%, and ($x_{10}$, $x_{11}$) from ($x_{8-11}$) with a probability of 50%. We concatenate the sampled tokens ($x_0, x_4, x_{10}, x_{11}$) with the target tokens and preserve their positions to predict the target tokens.

the attention distribution of the Transformer network empirically for modeling long-sequence dependency. Based on these observations, we raise the following question:

*Is it possible to simulate attention sparsity without modifying the architecture during the training?*

Therefore, in this paper, we introduce SPARSE TRAINING, which aims to extend the context window of existing LLM frameworks by leveraging continual pre-training within a fixed window size. Specifically, we argue that distant tokens generally provide less information for a token prediction compared to tokens that are closer to the target. In other words, most of computations (i.e., dot product) between distant tokens and the target tokens are redundant. Hence, the core idea behind SPARSE TRAINING is to sample tokens from the distant tokens and simultaneously keep their corresponding positions, and then adopt the standard next-token prediction for the target tokens. This process is illustrated in Figure 1. More specifically, we divide the input sequences as the *memory* part and the *target* part. Based on the posterior distribution of attention sparsity, we devise a sampling policy over the memory part with a decay factor across the distance to collect tokens. That implies tokens closer to the target part will be sampled at a higher probability while the farther tokens are sampled at a lower probability. This design enables us to replicate the sparsity of long-sequence dependencies at the input level, rather than architecture. Generally, it also offers us three key benefits to model long-sequence dependency: 1) *Efficient Long-Sequence Training*. By training on the sampled sequence where the length $L_{sample} < L$, our method can reduce the space and time complexity from $O(L^2)$ to $O(L^2_{sample})$ when compared with directly training long sequences (Fu et al., 2024) on the Transformer network; 2) *Sparsity Simulation*. By applying a decay sampling policy across the length, our method also simulates the situation of the attention sparsity in long-sequence modeling; 3) *Architecture Invariance*. Compared with previous sparse architectures, SPARSE TRAINING does not involve any modifications to the architecture, which makes it adaptable to any LLM framework to extend its capability in modeling long-sequence dependency.

To verify the effectiveness of SPARSE TRAINING, we conduct extensive experiments on and public benchmark datasets. Experimental results demonstrate that by deploying SPARSE TRAINING over existing LLM frameworks, it can effectively improve the model's capability to infer over long contexts. Our contributions can be summarized as follows:

- We conduct an in-depth analysis of the statistical attention patterns in Transformers across different LLMs, and summarize several laws regarding attention distribution, including its sparsity, weight allocation and decay over distance.

- Based on our analysis, we propose SPARSE TRAINING, a novel training approach to extend context window size of LLMs, without any modifications in the architectures.

- Empirically, we demonstrate the effectiveness of SPARSE TRAINING through extensive experiments on multiple state-of-the-art LLMs over public benchmarks.

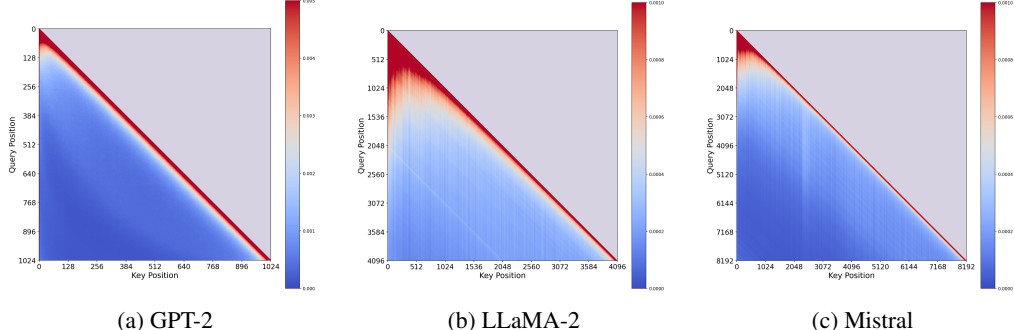

(a) GPT-2          (b) LLaMA-2          (c) Mistral

Figure 2: Attention visualization on different LLMs. GPT-2 is over 1024 samples with a length of 1024, LLaMA-2 is over 4096 samples with a length of 4096, and Mistral-7B is over 2048 samples with a length of 8192. All results are computed by averaging across samples and layers.

## 2 STATISTICAL LAWS OF ATTENTION PATTERNS

To unveil the secrets of sparsity beneath the attention mechanism of Transformer networks, we first analyze several statistical patterns of attention across different samples in this section. Here, in the standard Transformer architecture (Vaswani et al., 2017), the token features are aggregated through the self-attention mechanism as follows:

$$\widetilde{\mathbf{H}} = \text{Attn}_{\boldsymbol{\theta}_a}(\mathbf{H}) := \mathbf{H} + \frac{1}{N} \sum_{m=1}^{M} (\mathbf{V}_m \mathbf{H}) \times \sigma\big((\mathbf{Q}_m \mathbf{H})^\top (\mathbf{K}_m \mathbf{H})\big) \in \mathbb{R}^{D \times N} \tag{1}$$

where $\mathbf{H} \in \mathbb{R}^{D \times N}$ is the input sequence embedding and $\boldsymbol{\theta}_a = \{(\mathbf{V}_m, \mathbf{Q}_m, \mathbf{K}_m)\}_{m \in [M]} \subset \mathbb{R}^{D \times D}$ denotes the parameters with $M$ heads. $N$ is the number of input tokens and $D$ is the embedding dimension. $\sigma$ denotes the attention mask and activation, e.g., scaling by $\frac{1}{\sqrt{D}}$ followed by softmax operation. Conventionally, "attention matrix" refers to the matrix $\sigma\big((\mathbf{Q}_m \mathbf{H})^\top (\mathbf{K}_m \mathbf{H})\big) \in \mathbb{R}^{N \times N}$ with triangular masking applied, i.e., each token attends to all preceding tokens.

To better understand the attention patterns from a statistical viewpoint, we visualize the attention matrix across different LLMs (e.g., GPT-2 (Radford et al., 2019), LLama-2 (Touvron et al., 2023b) and Mistral (Jiang et al., 2023)) by calculating its average attention weights over each layer and sample, shown in Figure 2. All results are tested on the WikiText-103 dataset (Merity et al., 2017) and measured by the maximum length of their context window. Let $\mathbf{A}_{\mathcal{M}}$ denote the average attention matrix for language model $\mathcal{M}$. We discuss several key insights in the following subsections.

### 2.1 PARETO PRINCIPLE OF TRANSFORMERS

Generally, a common observation is that attention distribution always exhibits sparsity when processing long sequences. From Figure 2, we can clearly observe that the tokens close to the query tokens (i.e., diagonal red pixels) usually receive more attention than distant tokens. To further analyze the attention distribution, we also count the cumulative sum $S_k = \sum_{i=1}^{k} \alpha_{(i)}$ of the attention weight sorting by their distance to the query token or their ranked corresponding weight [1]. Our results are displayed in Figure 3. From Figure 3b, we can find that approximately 25% of the tokens account for the vast majority of the total attention, which we refer to as the "Pareto Principle [2] of Transformers". These observations also suggest that for long-sequence modeling, attention patterns are usually sparse and most of pair-wise computations in the attention operations are redundant. Our studies raise a question: is it possible to sample a few tokens for long-sequence modeling while simultaneously preserving such a sparsity?

### 2.2 ATTENTION DECAY WITH RELATIVE DISTANCES

Figure 3c presents the attention weight sum per 1024 tokens. From this Figure, the first bin contributes to over 50% percent of the total attention weight. Additionally, there is a clear descending

---

[1] We rank each token $x_i$ by their attention weight to guarantee its attention weight $\alpha_{(i)} \geq \alpha_{(i+1)}$.

[2] The original Pareto Principle from economics states that a small proportion of factors often account for a large portion of the effect. https://en.wikipedia.org/wiki/Pareto_principle.

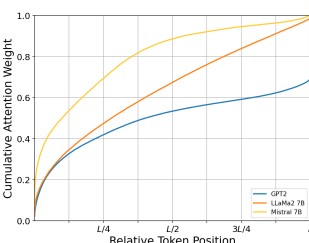
(a) Cumulative Sum (distance).

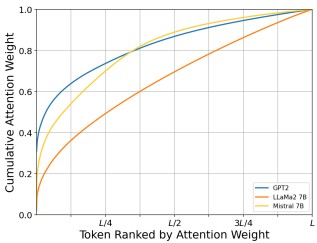
(b) Cumulative Sum (weight).

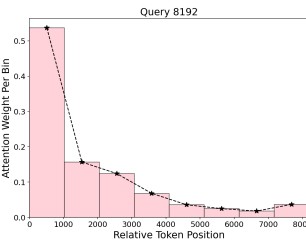
(c) Attention Sums Per Bin.

Figure 3: Spatial distribution of Attention in the Transformer network. (a) The cumulative sum of attention weight of each position; (b) The cumulative sum of attention weight sorted by the weight of each token in descending order; (c) We count the distribution of attention weight and divide it into bins where each bin includes 1024 tokens.

trend as the position increases except the last one [3]. In contrast, standard Transformer networks assume that each position contributes equally when calculating the outputs of attention layers, ignoring these evident statistical patterns. Therefore, we deem it important to incorporate such an attention decay to reduce the complexity of long-sequence modeling.

### 2.3 SPARSE ATTENTION IS NOT ALL YOU NEED

Inspired by the Pareto Principle in Transformers, some works (Xiao et al., 2024; Han et al., 2024; Jiang et al., 2023) explore applying some specific attention patterns to sample tokens for inference. They attribute attention distributions to two common patterns: sliding window and $\Lambda$-shape. The former only passes close tokens to Transformers, while the latter considers the first few tokens together with the close tokens critical to making predictions. However, as shown in Figure 3(a), the middle tokens (approximately from $L/4$ to the end) account for at least $30\%$ of the attention, indicating that these tokens may encode crucial information for downstream tasks. Moreover, these works do not adequately extend the capability of LLMs to achieve long-sequence dependency. Therefore, how to generalize existing LLM frameworks to unseen length via training still needs to be addressed.

## 3 SPARSE TRAINING

As mentioned previously, the backbone of most modern LLM frameworks is decoder-only Transformer, whose quadratic complexity in computing $(\mathbf{Q}_m\mathbf{H})^\top(\mathbf{K}_m\mathbf{H}) \in \mathbb{R}^{N \times N}$ in equation 1 makes it inefficient when handling long sequences (large $N$). To this end, we believe that an ideal solution to extend the capability of LLMs to generalize long sequences should meet these criteria:

- It should not introduce any modification over architectures to preserve its architectural integrity;
- It should be able to simulate the sparsity of the attention distribution in sequence computations;
- It should effectively reduce the time and space complexity, avoiding quadratic growth.

Therefore, in this paper, we introduce SPARSE TRAINING, a novel training strategy to extend existing LLMs to support long sequence generalization. The details are described below.

### 3.1 FRAMEWORK

Assume the final part of a long sequence as $X = \{x_{m+1}, \ldots, x_N\}$, where m starts from a large position (e.g., beyond 4096 in LLaMA-2). The conventional method to establish long-sequence training is to directly calculate the whole sequence from position 1 to N via attention operations (i.e., $O(N^2)$), while bringing massive and redundant computations. Therefore, we claim that the core challenge to address the long-context issue is how to bridge the connection between two distant tokens. However, considering the sparsity between the distant and the target tokens, we argue that not all pairwise computations in attention are essential, and some distant tokens could be ignored for modeling long contexts to simulate sparsity.

---

[3]Based on previous experiences, Transformer networks suffer from "attention sink" (Xiao et al., 2024) that means the first few tokens usually occupy a ratio of attention weight.

To this end, for an input sequence $X = \{x_1, \ldots, x_N\}$, we divide it into the memory part $X_{mem} = \{x_1, \ldots, x_m\}$ and the target part $X_{target} = \{x_{m+1}, \ldots, x_N\}$, where m exceeds the predefined context window L (e.g., 4096 in LLaMA-2) of original LLMs. Here, we assume $|N - m|$ is equal to $L/2$. Therefore, we propose SPARSE TRAINING, which aims to sample a sub-sequence $\tilde{X}_{mem} = \{\tilde{x}_{i_1}, \ldots, \tilde{x}_{i_{L/2}}\}$ from the memory part $X_{mem}$, where the sampled indices $\{i_1, \ldots, i_{L/2}\}$ are from $[1, m]$. Then, we concatenate the sampled $\tilde{X}_{mem}$ and the target part $X_{target}$ as the input sequence and thus employ the standard next-token prediction for the target part. To identify the long-range dependencies among sequences, we also preserve the corresponding positional indices of each token, as Transformer is a position-independent architecture [4]. Here, we use cross-entropy loss to optimize our model, and the objective function of SPARSE TRAINING is defined as:

$$\mathcal{L}_{\text{SparseTraining}}(X_{target}|\tilde{X}_{mem}, \theta) = -\frac{1}{|N - m|} \sum_{i=m+1}^{N} \log p(x_i|x_{m+1 \leq t < i}, \tilde{X}_{mem}, \theta), \quad (2)$$

Here, we enable the target part to follow the standard next-token prediction for modeling continuous sequences, and then we use the sampled memory part to establish the long-sequence dependencies between the target part and the distant tokens. Figure 1 also illustrates the pipeline of our method. In Figure 1, we sample four tokens $(x_0, x_4, x_{10}, x_{11})$ from the memory part, and then auto-regressively predict tokens in the target part. So, in this case, we extend the window size of the language model to 16 tokens while its predefined window size is 8. Therefore, in SPARSE TRAINING, its complexity is independent of the input sequence length $N$, stated as follows:

**Lemma 3.1** *Given length-$N$ sequences and an LLM pretrained on length $L < N$, SPARSE TRAINING reduces causal language modeling complexity from $O(N^2)$ to $O(L^2)$ for both space and time.*

We can find that this design enables us to conduct long-sequence training without any architectural modifications, and only requires $O(L^2)$ complexity during the training. In addition, we also design two techniques to enhance our model: 1) Sparse Sampling with decay over the distance to simulate attention sparsity in long-sequence dependency; 2) Mixed Training to guarantee the original capability of LLMs when $i \leq L$. More details are described below.

---

**Algorithm 1:** Sparse Sampling with Decay

**Require:** uniform(l, r, n) means uniformly sample n distinct tokens from position l to r.
**Input:** The Length of Memory Part M, The Number of Sample Tokens N, The Initial Sample Window W (default as N), The number of decay iterations T

```
1 def SparseSampling(M, N, W, T):
2     if M < 2W or T == 1 then
3         ids = uniform(1, M, N)
4     else
5         ids = uniform(M - W, M, N/2)
6         ids = concat(ids, SparseSampling(M − W, N/2, 2W, T − 1))
7     return ids
```

---

### 3.2 SPARSE SAMPLING WITH DECAY

SPARSE TRAINING adopts a sampling policy to sample distant tokens and build their connections with the target part. Based on our analysis in section 2, the attention distribution also manifests sparsity with the increasing distance. Therefore, using uniform sampling from the memory parts is unsuitable as it cannot highlight this characteristic. Consequently, we expect to develop a sparse sampling policy that should satisfy these two criteria: 1) Captures the sparsity of the attention distribution, ensuring sufficient allocation to nearby tokens that are likely to be important; 2) Reflects the decay pattern of attention with increasing relative distances. To this end, we design a sparse sampling with a decay over the distance, which is depicted in Algorithm 1. In our algorithm, we involve an initial window size W for sampling. If the length of memory part is smaller than twice the size of W, we employ a uniform sampling to obtain N tokens from the position 1 to M (Line 3). Otherwise, we uniformly sample $\frac{N}{2}$ tokens from the position $M - W$ to M (i.e., the closest interval

---

[4]Transformer identifies the order of tokens via their positional embeddings.

to the target part), and another $\frac{N}{2}$ tokens are sampled from the remaining memory part with a larger window (Lines 5-6). This design enables us to sample more tokens within the nearest window, but also guarantee that the farthest tokens can also be accessed. We also give some examples of our sampling policy in the Appendix A.1.

### 3.3 MIXED TRAINING

While our proposed SPARSE TRAINING can effectively help us capture long-range dependencies of the distant tokens, it will also suffer from another common issue: catastrophic forgetting (Luo et al., 2023; Wu et al., 2024; Kotha et al., 2024; Huang et al., 2024) in the original positions (i.e., From 1 to L). To address this issue, we devise mixed training that combines SPARSE TRAINING and standard next-token prediction on the original window to preserve the capability of LLMs in processing tokens within the position from 1 to L.

$$\mathcal{L} = \mathbb{E}[\sum_{i=m+1}^{N} \log p(\mathrm{x_i}|\mathrm{x_{m+1 \leq t < i}}, \tilde{X}_{\mathrm{mem}}, \theta)] + \beta \mathbb{E}[\sum_{i=1}^{L} \log p(\mathrm{x_i}|\mathrm{x_{t < i}}, \theta)], \quad (3)$$

where $\beta$ is a hyper-parameter to balance the sparse training and the original next-token prediction, empirically set to 1. Specifically, we only tune $\mathbf{Q}, \mathbf{K}$ of each Transformer layer [5] to further reduce computations and also preserve original knowledge. Following previous experiences (Ouyang et al., 2022; Ziegler et al., 2019; Dong et al., 2023), this simple technique can effectively guarantee the model does not deviate significantly from the original pre-trained one during the continual training.

### 3.4 DISCUSSION

In this section, we also want to discuss why SPARSE TRAINING is effective at processing long-context information. We attribute its effectiveness from two perspectives as follows:

**Positional Generalization** The critical part of attention operation to capture dependency is $(\mathbf{Q}_m \mathbf{H})^\top (\mathbf{K}_m \mathbf{H})$, when $\mathbf{Q}_m$ and $\mathbf{K}_m$ have been applied with positional information. Therefore, the way to enable model to learn positional information beyond the original context window is important. During pre-training with window size $L$, the model only accessed the positional encoding of positions $(1, \ldots, L)$, and thus cannot be generalized to untrained positional encoding. However, in SPARSE TRAINING, we enable model to access more positions beyond $L$ for optimization.

**Lemma 3.2** *With the sampling strategy described in 3.2, each position $n$ of the input sequence has a non-zero probability of being sampled, and such probability generally decays by distance.*

**Training Mismatch** Another issue of SPARSE TRAINING is whether it can build next-token prediction based on the sampled memory tokens. We deem that SPARSE TRAINING can be considered as a kind of dropout (Srivastava et al., 2014) at the token level, compared with standard training. That makes it compatible with other LLM training techniques and does not involve any modification at the architecture level.

## 4 EXPERIMENT

We evaluate the effectiveness of SPARSE TRAINING to extend the context window of Transformer networks via the continual training. We conduct a series of experiments using the LLaMA-2-7B model [6] (Touvron et al., 2023b) with a pre-trained context window of 4096. In particular, we aim to study the following research questions: **RQ1**: How effective is our SPARSE TRAINING at extending the context window of a given large language model? **RQ2**: As a training technique, will SPARSE TRAINING preserve the language ability acquired during pre-training? **RQ3**: Can SPARSE TRAINING reduce the computational complexity when modeling long contexts, as stated in Lemma 3.1? **RQ4**: SPARSE TRAINING has two components, the crucial SPARSE TRAINING itself and the mixed training. Is mixed training contributing to the overall effectiveness? All experiments are conducted on an Ubuntu server with 8 Nvidia H100 GPUs, each with 80GB of graphic memory.

---

[5] For most of LLM frameworks, they apply RoPE (Su et al., 2024) to query and key vectors.

[6] Model weights are available at `https://huggingface.co/meta-llama/Llama-2-7b-hf`.

Table 1: Perplexity (↓) and Accuracy (↑) of LLaMA-2-7B on several datasets. The performance of LLaMA-2-7B after SPARSE TRAINING is stable and improves with longer contexts.

| Model | Context Length | PG19 | | arXiv | | SlimPajama | |
|---|---|---|---|---|---|---|---|
| | | PPL (↓) | Acc (↑) | PPL (↓) | Acc (↑) | PPL (↓) | Acc (↑) |
| Vanilla | 4K | 7.88 | 0.54 | 8.22 | 0.54 | 5.73 | 0.61 |
| | 8K | 151.83 | 0.31 | 140.32 | 0.32 | 130.07 | 0.34 |
| | 16K | 1052.86 | 0.15 | 1209.21 | 0.16 | 1269.29 | 0.17 |
| | 32K | 2638.58 | 0.08 | 3417.44 | 0.08 | 2584.39 | 0.1 |
| | 64K | 5438.16 | 0.05 | 7154.67 | 0.04 | 6172.95 | 0.05 |
| Sparse Training | 8K | 11.08 | 0.48 | 16.77 | 0.45 | 13.43 | 0.48 |
| | 16K | 9.59 | 0.51 | 13.76 | 0.48 | 10.69 | 0.51 |
| | 32K | 8.48 | 0.53 | 9.62 | 0.52 | 7.90 | 0.55 |
| | 64K | 8.02 | 0.54 | 9.15 | 0.53 | 7.39 | 0.57 |

*Training*. We use the LLaMA-2-7B model as the backbone network and continue to train it on the PG19 (Rae et al., 2020) dataset. We adopt the training techniques described in Section 3.3 to prevent catastrophic forgetting. This results in approximately one billion trainable parameters ($\sim 13\%$ of all parameters). To further optimize the GPU memory usage, we leverage Huggingface Accelerate (Gugger et al., 2022) plus Deepspeed (Rajbhandari et al., 2020), speed up with Zerostage 2 by using BFloat16. For every 1,000 steps, we extend the context window by 2K, allowing us to gradually increase LLaMA-2-7B's context window from 4K to 64K. Because the complexity of SPARSE TRAINING does not depend on the input sequence length (Lemma 3.1), each 1000 steps take approximately 30 minutes and the whole training can be done in less than 16 hours. The training curves are provided in Appendix C.3 and more details can be found in Appendix C.

*Evaluation*. For PG19 (Rae et al., 2020), we select a ratio of 5% of this dataset as a basic sanity test. Then, to validate that SPARSE TRAINING empowers language model with general long-range dependency, we also adopt arXiv (Clement et al., 2019) and Slimpajama (Soboleva et al., 2023) to measure the long-context capability of our trained model. Here, we mainly report results by perplexity and accuracy. Then, we also adopt LongBench (Bai et al., 2024), a multi-task long-context benchmark, to evaluate performance over 12 datasets of 6 downstream tasks. The details of datasets can be found in Appendix B.

## 4.1 EXTENDING CONTEXT WINDOW WITH SPARSE TRAINING (RQ1)

In this subsection, we investigate the effectiveness of SPARSE TRAINING to extend the context window of a given large language model. Here, we evaluate our method on the PG19, arXiv and SlimPajama, using LLaMA-2-7B model with SPARSE TRAINING. Besides, we also evaluate the vanilla LLaMA-2-7B model for comparison. The results are reported in Table 1. The results show that while the vanilla model has limited performance on sequences beyond its original context window, SPARSE TRAINING can significantly improve long-context capability of LLMs, demonstrated by stable perplexity and accuracy close to vanilla LLaMA-2-7B on 4K sequences. Moreover, as context length increases and perplexity decreases, SPARSE TRAINING can also enable the model to achieve the capability of learning long context in a right way. Besides, we can also observe significant improvement not only on PG19, but also on out-of-domain datasets (e.g., Arxiv and Slimpajama), proving that SPARSE TRAINING enhances robust generalization across varying sequence lengths. To further validate the generalization of our proposed method in processing long-sequence dependency, we conduct experiments on LongBench datasets, and the results are reported in Table 2. We find SPARSE TRAINING significantly improves the performance across all datasets under each downstream category, which shares a similar conclusion above. Besides, we measure the perplexity on LongBench, reported in Appendix C.4.

To further understand the mechanism of our method in learning long context, we visualize the average attention weights on the LongBench dataset, to compare our method with vanilla model. As shown in Figure 4, we find that the attention distribution of the vanilla model is highly concentrated on the initial few tokens and some specific positions beyond the context window, leading to failure in handling long sequences. In contrast, our method demonstrate a smooth attention distribution over a longer context window, which indicates our method can better capture long-sequence dependencies.

Table 2: Accuracy (↑) of LLaMA-2-7B on LongBench datasets. The performance of LLaMA-2-7B after SPARSE TRAINING is stable and slightly improves with longer contexts.

| Model | Context Length | Single-Doc QA | | Multi-Doc QA | | Summarization | |
|---|---|---|---|---|---|---|---|
| | | Qasper | MultiFieldQA | HotPotQA | WikiMQA | GovReport | MultiNews |
| Vanilla | 8K | 0.28 | 0.37 | 0.34 | 0.33 | 0.35 | 0.34 |
| | 16K | 0.14 | 0.17 | 0.17 | 0.17 | 0.18 | 0.18 |
| | 32K | 0.08 | 0.09 | 0.09 | 0.09 | 0.09 | 0.09 |
| | 64K | 0.04 | 0.05 | 0.04 | 0.05 | 0.04 | 0.05 |
| Sparse Training | 8K | 0.47 | 0.49 | 0.51 | 0.50 | 0.51 | 0.53 |
| | 16K | 0.49 | 0.53 | 0.54 | 0.54 | 0.53 | 0.54 |
| | 32K | 0.52 | 0.59 | 0.58 | 0.58 | 0.55 | 0.57 |
| | 64K | 0.54 | 0.61 | 0.60 | 0.59 | 0.56 | 0.56 |

| Model | Context Length | Few-shot Learning | | Synthetic Task | | Code Completion | |
|---|---|---|---|---|---|---|---|
| | | TREC | TriviaQA | PassageCount | PassageRetrieval | LCC | RepoBench-P |
| Vanilla | 8K | 0.40 | 0.54 | 0.47 | 0.46 | 0.51 | 0.50 |
| | 16K | 0.28 | 0.32 | 0.32 | 0.31 | 0.34 | 0.34 |
| | 32K | 0.15 | 0.16 | 0.17 | 0.16 | 0.17 | 0.17 |
| | 64K | 0.05 | 0.05 | 0.05 | 0.04 | 0.06 | 0.07 |
| Sparse Training | 8K | 0.61 | 0.50 | 0.52 | 0.46 | 0.66 | 0.65 |
| | 16K | 0.63 | 0.54 | 0.55 | 0.47 | 0.67 | 0.65 |
| | 32K | 0.67 | 0.58 | 0.56 | 0.50 | 0.78 | 0.79 |
| | 64K | 0.68 | 0.59 | 0.58 | 0.52 | 0.81 | 0.81 |

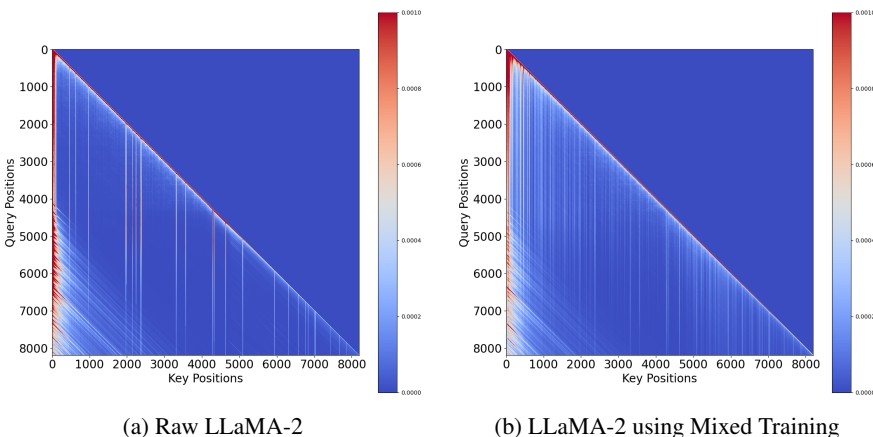

(a) Raw LLaMA-2  (b) LLaMA-2 using Mixed Training

Figure 4: Attention visualization on LLaMA 2 after SPARSE TRAINING Qasper Task from Long-bench. The results are computed by averaging across different samples, heads, and layers.

## 4.2 MAINTAINING PRE-TRAINED LANGUAGE MODELING ABILITY (RQ2)

As aforementioned in Section 3.3, we also need to ensure the capability of language models to process tokens within the original context window. There-fore, in this part, we conduct experiments to validate our method and vanilla model in evaluating the con-text window with 4K tokens. We report our results on PG19, arXiv, SlimPajama in Table 4, and Long-Bench in Table 3. From the results, we can find that textscSparse Training configured with mixed train-ing can achieve similar performance when compared to the vanilla model in different settings, which also demonstrates the effectiveness of our design in preserving the original knowledge of language models.

Table 4: Perplexity (↓) and Accuracy (↑) on several datasets with 4K input length.

| Model | Metric | PG19 | arXiv | SlimPajama |
|---|---|---|---|---|
| Vanilla | PPL (↓) | 7.88 | 8.22 | 5.73 |
| | Acc (↑) | 0.54 | 0.54 | 0.61 |
| Sparse Training | PPL (↓) | 7.90 | 8.36 | 5.89 |
| | Acc (↓) | 0.54 | 0.54 | 0.6 |

Table 3: Performance on LongBench datasets with 4K input length. The performance of LLaMA-2-7B after SPARSE TRAINING on 4K (pre-train window length) is close to the vanilla model.

| Model | Metric | Single-Doc QA | | Multi-Doc QA | | Summarization | |
|-------|--------|--------|--------------|----------|---------|-----------|-----------|
| | | Qasper | MultiFieldQA | HotPotQA | WikiMQA | GovReport | MultiNews |
| Vanilla | PPL ($\downarrow$) | 6.99 | 5.17 | 4.98 | 5.27 | 4.71 | 4.47 |
| | Acc ($\uparrow$) | 0.56 | 0.62 | 0.63 | 0.61 | 0.63 | 0.65 |
| Sparse Training | PPL ($\downarrow$) | 7.12 | 5.51 | 5.27 | 5.50 | 4.83 | 4.57 |
| | Acc ($\uparrow$) | 0.56 | 0.61 | 0.62 | 0.61 | 0.62 | 0.65 |
| Model | Metric | Few-shot Learning | | Synthetic Task | | Code Completion | |
| | | TREC | TriviaQA | PassageCount | PassageRetrieval | LCC | RepoBench-P |
| Vanilla | PPL ($\downarrow$) | 4.97 | 5.18 | 4.12 | 7.39 | 2.09 | 2.05 |
| | Acc ($\uparrow$) | 0.69 | 0.62 | 0.69 | 0.56 | 0.83 | 0.83 |
| Sparse Training | PPL ($\downarrow$) | 5.14 | 6.21 | 4.65 | 7.86 | 2.13 | 2.07 |
| | Acc ($\uparrow$) | 0.68 | 0.60 | 0.66 | 0.55 | 0.83 | 0.83 |

## 4.3 REDUCING LONG-CONTEXT TRAINING COMPLEXITY (RQ3)

As mentioned above, by sampling a ratio of the memory part, we can extend long-sequence training with quadratic complexity for a fixed length, and thus reduce both space and time complexity. Here, we respectively extend the context window from 4K to 8K, 16K, 32K, and 64K, and then report the time consumption per step in Table 5. From Table 5, we observe that SPARSE TRAINING can achieve similar time cost compared to standard training under 4K contexts. When we scale up the context length, our method can still guarantee

Table 5: Time consumption (seconds per step) training LLaMA-2-7B on PG19. OOM: out of GPU memory.

| Training Scheme | 4K | 8K | 16K | 32K | 64K |
|-----------------|------|------|------|------|------|
| Standard | 1.56 | 3.63 | OOM | OOM | OOM |
| Sparse Training | - | 1.57 | 1.57 | 1.58 | 1.58 |

that our time consumption is independent of the fixed input length. Moreover, our method can avoid GPU memory explosion, while the standard method suffers from out-of-memory (OOM) issue when training on longer sequences.

## 4.4 ABLATION STUDY (RQ4)

In this subsection, we further validate the effectiveness of mixed training in our design. Here, we train another LLaMA-2 model by using SPARSE TRAINING, but without the mixed training. We evaluate the model on LongBench, and the results are reported in Table 6. We observe that the results are worse than using our proposed SPARSE TRAINING with mixed training from 16K to 64K, especially in 16K and 32K. We claim that this performance degradation is caused by catastrophic forgetting. Overall, these results also demonstrate the necessity of mixed training in our method. More results can refer to Appendix C.5.

Table 6: PPL ($\downarrow$) of SPARSE TRAINING without mixed training on LongBench.

| Model | Context Length | Single-Doc QA | | Multi-Doc QA | | Summarization | |
|-------|----------------|--------|--------------|----------|---------|-----------|-----------|
| | | Qasper | MultiFieldQA | HotPotQA | WikiMQA | GovReport | MultiNews |
| Sparse Training w/o Mixed Training | 16K | 1332.19 | 1134.13 | 1806.22 | 1933.51 | 751.38 | 1005.19 |
| | 32K | 1471.31 | 1269.43 | 2870.13 | 4162.76 | 5829.82 | 5470.50 |
| | 64K | 310.02 | 368.17 | 433.99 | 418.16 | 608.48 | 1135.98 |

## 5 RELATED WORK

With the rise of advanced LLMs , how to extend the capability of Transformer-based LLMs to generalize across long sequences has become an ongoing challenge. Generally, the current approaches to generalize the context window of LLMs can be grouped into two categories, which are as follows:

**Efficient Training with Sparse Architectures** The standard complexity of Transformer networks is known to scale as $O(L^2)$. To alleviate the burden of quadratic complexity, many research efforts (Tay et al., 2023) have focused on developing advanced or sparse architectures to effectively approximate the attention mechanism. Specifically, some works like Sparse Transformer (Child et al., 2019) apply sparse factorization to the attention matrix, thus reduce the complexity to $O(L\sqrt{L})$. Some other works (e.g., Linformer (Wang et al., 2020) and Performer Choromanski et al. (2021)) attempt to approximate the self-attention matrix via low-rank decomposition. Besides, some works (e.g., Reformer (Kitaev et al., 2020), Block-wise Self-Attention (Qiu et al., 2020), LongFormer (Beltagy et al., 2020), Big Bird (Zaheer et al., 2020), LongNet (Ding et al., 2023)) propose some fixed sparse attention patterns to reduce time complexity. Recently, some papers have attempted to develop parallelized RNN to address this problem, like Mamba (Gu & Dao, 2023), RWKV (Peng et al., 2023) and RetNet (Sun et al., 2023a). In order to extend the context window of Transformer, several methods explore the use of hybrid window-full attention for training, that means some layers adopt full attention while other use sparse attention patterns. For example, Long Llama (Tworkowski et al., 2023) uses the bottom layers to retrieve the most relevant top-k tokens, then performs attention operations on these tokens to reduce computational complexity. However, the scalability and capability of these works are still beneath fully attention architectures, and thus most mainstream LLM frameworks still adopt standard Transformer architecture (i.e., full attention) as the backbone network. Compared with these works, SPARSE TRAINING does not involve any modifications over architectures but simulates sparsity at the input-level. Therefore, it can also be considered as a post-training technique that can be adopted to existing LLM frameworks, and maintain the complexity within a fixed window size.

**Extend Context Window with Length Extrapolation** Instead of directly using sparse architecture, a large amount of research focuses on inferring unseen length beyond the pre-training window size based on the original Transformer network. These works can be considered as a kind of position engineering (Zhao et al., 2023). Among these works, RoPE (Su et al., 2024) and Alibi (Press et al., 2022) are the most representative ones. These works can effectively encode relative positional information without any learnable parameters, allowing for length extrapolation. Building on this, some other works (CAPE (Likhomanenko et al., 2021), SANDWICH (Chi et al., 2023), xPOS (Sun et al., 2023b), LongRoPE (Ding et al., 2024), NoPE (Kazemnejad et al., 2023), FIRE (Li et al., 2024), and CLEX (Chen et al., 2024)) also extend different positional encoding. However, as models have not been generalized to unseen positions through training, these works still suffer from performance degradation. Therefore, some works propose position interpolation, that re-scales the out-of-distribution positional encoding within the pre-trained window size (Chen et al., 2023). YaRN (Peng et al., 2024) leverages neural tangent kernel (NTK) to interpolate RoPE and generalize LLaMA-2 to support 128K tokens. Besides, a similar work (Ruoss et al., 2023) introduces to randomly sample some tokens to extend length generalization but ignores the sparsity when modeling long-sequence dependency. Generally, our method is orthogonal to these method as we aim to generalize the long-sequence capability of LLMs from the training level. SPARSE TRAINING can also use these advanced positional embeddings to encode long sequences, while in this paper, we mainly use RoPE as the backbone for experiments.

## 6 CONCLUSION

In this paper, we present a novel training framework that can efficiently extend the context window of LLM frameworks based on the Transformer architecture, named SPARSE TRAINING. Specifically, we first analyze statistical laws of existing attention patterns and identify the phenomenon of "Pareto Principle of Transformer". Based on these observations, we introduce SPARSE TRAINING, which employs a sampling policy with a decay factor across the distance to gather tokens as the conditional part for long-sequence prediction. Based on the sampled tokens with their corresponding positions, we can directly adopt the standard next-token prediction for the long sequences. Benefiting from such a design, our method can effectively extend the context window of LLM frameworks within a fixed window training. In addition, compared with previous sparse architectures, SPARSE TRAINING will not introduce any modification over the architecture, and can also simulate the attention sparsity at the input level. Experimental results also demonstrate the effectiveness of our proposed method in processing long-sequence dependency.

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

# A    TECHNICAL DETAILS AND ANALYSIS

## A.1    EXAMPLE OF SPARSE SAMPLING WITH DECAY

In this part, we will present example of our design sparse sampling strategy with decay. We assume the window size as W, and then illustrate how our method allocates sampled tokens based on different length M of the memory part. The examples are presented in Table 7. We can find that when the length M of the memory part is just M, we will directly sample all tokens from this nearest window. When $M \in (W, 4W)$, we will sample $\frac{N}{2}$ tokens and the remaining $\frac{N}{2}$ tokens will be gather from the remaining windows, and so on. During our sampling, we also introduce the maximum number of decay iteration $T$ as when the sampling window is too distant, the influence of tokens within this range can be regarded as insignificant.

| Settings | Length | | | | | | | |
|---|---|---|---|---|---|---|---|---|
| | W | W | W | W | W | W | W | W |
| M = W | N | | | | | | | |
| M = 2W | $\frac{N}{2}$ | $\frac{N}{2}$ | | | | | | |
| M = 3W | $\frac{N}{2}$ | $\frac{N}{2}$ | | | | | | |
| M = 4W | $\frac{N}{2}$ | $\frac{N}{2}$ | | | | | | |
| M = 5W | $\frac{N}{2}$ | $\frac{N}{4}$ | $\frac{N}{4}$ | | | | | |
| M = 6W | $\frac{N}{2}$ | $\frac{N}{4}$ | $\frac{N}{4}$ | | | | | |
| M = 7W | $\frac{N}{2}$ | $\frac{N}{4}$ | $\frac{N}{4}$ | | | | | |

Table 7: Example of Sparse Sampling with decay.

## A.2    COMPLETE FORMULATION OF SPARSE TRAINING

In SPARSE TRAINING, we are given (1) post-training text corpus $\mathcal{C}$; (2) a pre-trained language model with vocabulary $\mathcal{V}$, embedding size $D$, pretrain context window length $L$ and parameters $\theta_{\mathcal{M}} = (\theta_{\mathsf{PE}}, \theta_{\mathsf{TE}}, \theta_{\mathsf{OUTPUT}}, \{\theta_{\mathsf{Attn}}^{(k)}, \theta_{\mathsf{Proj}}^{(k)}, \theta_{\mathsf{FF}}^{(k)}\}_{k=0}^{K-1})$, respectively for positional encoding, token embedding, the final linear output layer, and $K$ decoder layers. For each length-$N$ sequence $(N > L)$ of input tokens $X = (x_1, x_2, \ldots, x_N) \in \mathcal{V}^N$ in a data batch $\mathcal{B} = \{b_1, b_2, \ldots, b_{bsz}\}$ from the post-training corpus $\mathcal{C}$, SPARSE TRAINING samples a sub-sequence $X' = \mathsf{SAMPLE}(X) = (x_{i_1}, x_{i_2}, \ldots, x_{i_L}) \in \mathcal{V}^L$, with the sampled indices $(i_1, i_2, \ldots, i_L) \sim \mathsf{SAMPLE}(\cdot)$. Then, The token embedding $\theta_{\mathsf{TE}} \in \mathbb{R}^{D \times |\mathcal{V}|}$ maps the sequence to its embedding matrix $\theta_{\mathsf{TE}}[X'] \in \mathbb{R}^{L \times D}$. After that, $\theta_{\mathsf{PE}}$ calculates the positional encoding $\theta_{\mathsf{PE}}[i_1, i_2, \ldots, i_L] \in \mathbb{R}^{D \times L}$ and adds to $\theta_{\mathsf{TE}}[X']$ element-wise to obtain input sequence embedding as input of decoder layer 0.

$$\mathbf{H}'^{(0)} = \mathbf{H}' = \theta_{\mathsf{TE}}[X'] + \theta_{\mathsf{PE}}[i_1, i_2, \ldots, i_L] \in \mathbb{R}^{D \times L} \tag{4}$$

For each decoder layer $\{\theta_{\mathsf{Attn}}^{(k)}, \theta_{\mathsf{Proj}}^{(k)}, \theta_{\mathsf{FF}}^{(k)}\}$, $\theta_{\mathsf{Attn}}^{(k)} = \{(\mathbf{V}_m, \mathbf{Q}_m, \mathbf{K}_m)\}_{m \in [M]} \subset \mathbb{R}^{D \times D}$ computes $\widetilde{\mathbf{H}'}^{(k)} = \mathsf{Attn}_{\theta_{\mathsf{Attn}}^{(k)}}(\mathbf{H}'^{(k)}) \in \mathbb{R}^{D \times L}$ with value, query, key projections and attention mask as in Equation 1; $\theta_{\mathsf{Proj}}^{(k)} \in \mathbb{R}^{D \times D}$ projects $\widetilde{\mathbf{H}'}^{(k)}$ to attention output $\mathsf{Proj}_{\theta_{\mathsf{Proj}}^{(k)}}(\widetilde{\mathbf{H}'}^{(k)}) \in \mathbb{R}^{D \times L}$; the feedforward network further processes the attention output by adding residual, normalization and passing it through an MLP to obtain inputs of the next decoder layer.

$$\mathbf{H}'^{(k+1)} = \mathsf{FF}_{\theta_{\mathsf{FF}}^{(k)}}(\mathsf{Proj}_{\theta_{\mathsf{Proj}}^{(k)}}(\widetilde{\mathbf{H}'}^{(k)})) = \mathsf{FF}_{\theta_{\mathsf{FF}}^{(k)}}(\mathsf{Proj}_{\theta_{\mathsf{Proj}}^{(k)}}(\mathsf{Attn}_{\theta_{\mathsf{Attn}}^{(k)}}(\mathbf{H}'^{(k)}))) \in \mathbb{R}^{D \times L} \tag{5}$$

Let $\mathbf{H}'^{(K)} \in \mathbb{R}^{D \times L}$ denotes the output of the last decoder layer $K - 1$. Finally, an output layer with activation $\mathsf{OUTPUT}_{\theta_{\mathsf{OUTPUT}}} : \mathbb{R}^D \to \mathbb{R}^{|\mathcal{V}|}$ converts each token's hidden states to a probability distribution over vocabulary $\mathcal{V}$.

$$p_{\theta_{\mathcal{M}}}(\cdot \mid x_{i_1}, \ldots, x_{i_l}) = \mathsf{OUTPUT}_{\theta_{\mathsf{OUTPUT}}}(\mathbf{H}'^{(K)}[:, l]) \in \mathbb{R}^{|\mathcal{V}|} \tag{6}$$

where $\mathbf{H}'^{(K)}[:, l]$ denotes the $l^{th}$ column of $\mathbf{H}'^{(K)}$, i.e., the hidden states of the $l^{th}$ token. Define the Transformer function $\mathsf{TF}_{\theta_{\mathcal{M}}}(x_{i_1}, \ldots, x_{i_l}) \triangleq \arg\max_{y \in \mathcal{V}} p_{\theta_{\mathcal{M}}}(x_{i_1}, \ldots, x_{i_l})$, e.g., the token in $\mathcal{V}$ that maximizes $p_{\theta_{\mathcal{M}}}(\cdot \mid x_{i_1}, \ldots, x_{i_l})$. Following the standard auto-regressive next-token prediction language modeling, we maximize the probability of $\mathsf{TF}_{\theta_{\mathcal{M}}}(x_{i_1}, \ldots, x_{i_l}) = x_{i_{l+1}}$ by optimizing the cross-entropy loss between the true next token's distribution (one-hot) and $p_{\theta_{\mathcal{M}}}(\cdot \mid x_{i_1}, \ldots, x_{i_l})$.

$$\mathcal{L}_X(\theta_{\mathcal{M}}) = \mathcal{L}_{X'}(\theta_{\mathcal{M}}) = -\sum_{l=1}^{L-1} \log p_{\theta_{\mathcal{M}}}(x_{i_{l+1}} \mid x_{i_1} \ldots, x_{i_l}) \tag{7}$$

While the original $\mathbf{H}$ in Equation 1 is in $\mathbb{R}^{D \times N}$, by $\mathsf{SAMPLE}(\cdot)$, $\mathbf{H}' \in \mathbb{R}^{D \times L}$. As a direct result, the complexity of SPARSE TRAINING is independent of the input sequence length $N$, stated as in Lemma 3.1.

To mitigate the forgetting during SPARSE TRAINING, we adopt three techniques. (1) Only tune $\mathbf{Q}, \mathbf{K}$ and $\theta_{\mathsf{PE}}$ parameters. During pretraining, parameters that are unrelated to long-range dependency are already well-optimized. Only tuning parameters related to longer positions is beneficial for both pretrain knowledge preservation and efficiency. (2) Back-propagate with respect to the loss when predicting the target part only. While the memory part provides context for predicting the target part, it is not semantically continuous. Therefore, for each sequence $X$ in the post-training corpus $\mathcal{C}$, instead of Equation 7, we let the loss over it to be $\mathcal{L}_X(\theta_{\mathcal{M}}) = -\sum_{l=t}^{L-1} \log p_{\theta_{\mathcal{M}}}(x_{i_{l+1}} \mid x_{i_1}, \ldots, x_{i_l})$, where $t$ is the start of target part. (3) Regularize post-training with KL divergence to the original model is a common practice to ensure that the model does not deviate significantly from the original pre-trained one (Ouyang et al., 2022; Ziegler et al., 2019; Dong et al., 2023). We adopt the KL regularization in SPARSE TRAINING, leading to the following loss function, where $\mathsf{UNIFORM}(1, N)$ samples a random integer index between 1 and $N$, both inclusive. $X[i]$ and $X[:i]$ respectively denotes the $i^{th}$ token and the first $i$ tokens of $X$.

$$\mathcal{L}(\theta_{\mathcal{M}}) = -\mathbb{E}_{X \sim \mathcal{C}, \{i_j\}_{j=1}^L \sim \mathsf{SAMPLE}_{\mathcal{M}}(\cdot)}\Big[\sum_{l=t}^{L-1} \log p_{\theta_{\mathcal{M}}}(X[i_{l+1}] \mid X[i_1], \ldots, X[i_l])\Big]$$
$$+ \beta \mathbb{E}_{X \sim \mathcal{C}, l \sim \mathsf{UNIFORM}(1,N)} KL(p_{\theta_{\mathcal{M}}}(\cdot \mid X[:l]) || p_{\theta_{\mathcal{M}_0}}(\cdot \mid X[:l])) \tag{8}$$

where $\beta$ is a hyper-parameter to balance the sparse training and the original next-token prediction

### A.3 USE TOP-K ATTENTION IN GPT2 INFERENCE STEP

We have visualized the trend between the number of Top-K highest attention value used during the inference step of GPT2 and the value of two key performance metric, perplexity and accuracy, across multiple datasets from LongBench dataset (Bai et al., 2024). The results are illustrated in Figure 5. Despite the difference in values of perplexity and accuracy across different datasets, these figures still reveal a very clear and consistent trend: as the Top-K value increases, perplexity initially drops quickly, while accuracy sharply rises, before both metrics stabilize at higher K values. For all datasets, at very low K values (under 50), perplexity is high and accuracy is low, indicating that the model performs poorly due to limited access to relevant information. However, as K increases, perplexity undergoes a rapid decline, and accuracy improves sharply. In fact, the most significant changes in both metrics occur within the first few hundred K values. This trend suggests that a relatively small number of key-value pairs provide the majority of useful context for the model's predictions. Once K reaches around 500, the accuracy curve flattens and the perplexity plateaus, indicating that further increases in K yield limited improvement in the performance. The stabilization of both perplexity and accuracy across datasets highlights an underlying pattern of attention mechanisms: a limited number of Top-K weights is sufficient to capture most relevant information, making larger Top-K values computationally unnecessary beyond a threshold of approximately 500. A modest number of high-scoring key-value pairs capture the majority of relevant information needed for effective language modeling. After this optimal range is reached, further increases in K yield no significant improvements in either perplexity or accuracy, implying that the model has already captured the essential context.

### A.4 ATTENTION VISUALIZATION OF LLAMA2 MODEL

In this section, we present a visualization of attention weight distributions for the LLaMA2 models n the LongBench dataset, since understanding how attention weights are distributed across tokens

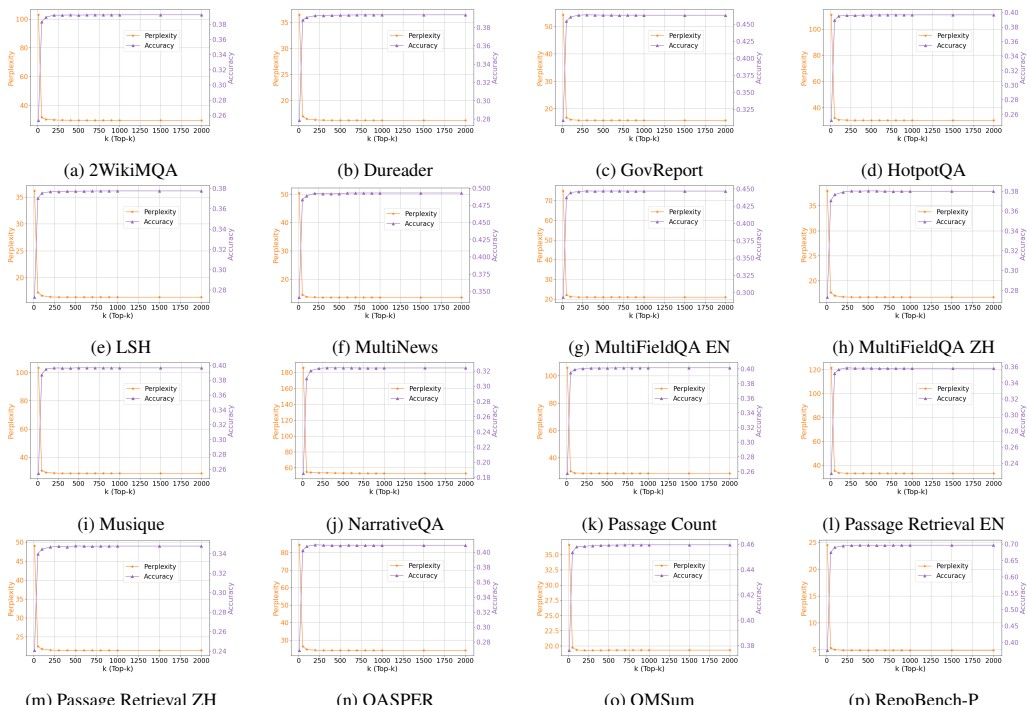

Figure 5: Top-k Perplexity and Accuracy Results for Various Datasets in Longbench

in long sequences provides insights into the models' behavior when dealing with large inputs. The model is evaluated in terms of cumulative attention weights across different query token positions in the last layer of each model. The results are illustrated in Figure 6 and Figure 7 .

One of the interesting phenomena that can be observed in the attention visualizations is "Pareto Principle of Transformers." This principle is an adaptation of the well-known Pareto distribution, which states that a small proportion of the causes is responsible for the majority of the effects. In the context of Transformer and attention mechanisms, the inherent sparsity of Transformer suggests that a large portion of attention weights is concentrated on a small fraction of key tokens when the sequence is long, while the majority of key tokens receive very little attention.

In long sequence modeling, such as the LongBench dataset, the Pareto Principle becomes evident. As demonstrated in the figures, a high percentage of attention weights tends to accumulate among a small subset of the highest-ranked key tokens. Notably, this phenomenon persists even after removing the tokens responsible for the "attention sink". In each subfigure, a larger number of key tokens contribute to the cumulative attention in the rescaled version, as the attention sink has been eliminated and fewer keys hold significant attention weights. This observation supports the notion that Transformers could benefit from our method by focusing on sparse key tokens. For sequences longer than 2000 tokens, the concentration of attention on a small set of tokens suggests the it is possible to employ methods like *SparseTraining* to significantly reduce computational complexity while preserving model performance.

## B  DATASET DETAILS

In the main experiments, we utilized a variety of datasets to validate the effectiveness of SPARSE TRAINING, including PG19, arXiv, SlimPajama, and 12 additional datasets from LongBench.

**PG19.**   PG19[7] includes a set of books extracted from the Project Gutenberg books library, that were published before 1919. It is significantly larger than previous benchmarks, with documents

---

[7] https://huggingface.co/datasets/deepmind/pg19

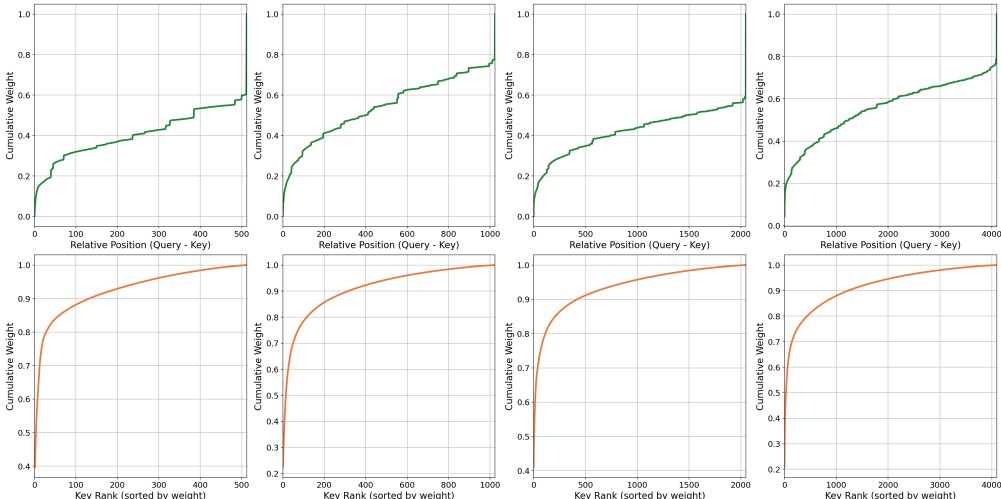

Figure 6: LLaMa2 cumulative attention weights in the last layer, visualized by both the relative distance between the query and key tokens, and by the key token rank (sorted by attention weight), with query positions at 512, 1024, 2048, and 4096 tokens, respectively.

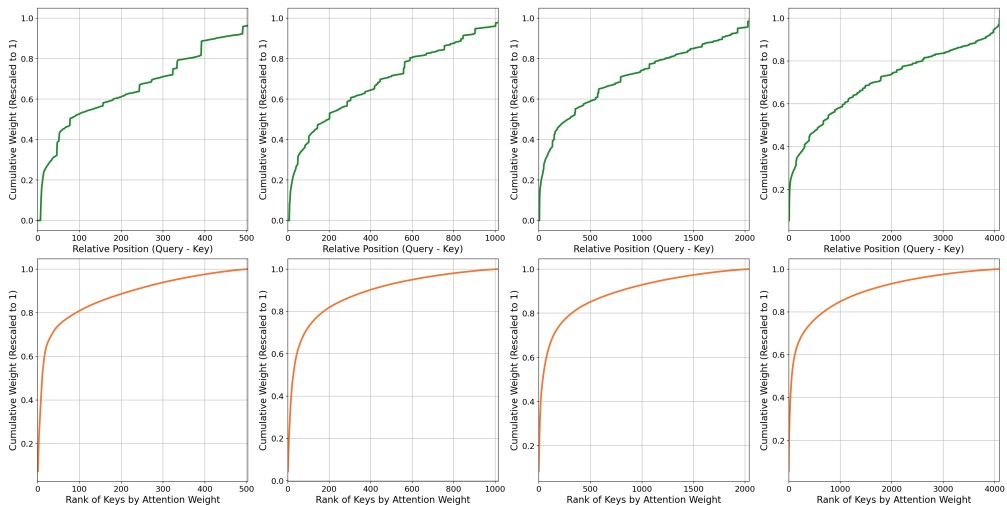

Figure 7: Rescaled LLaMA2 cumulative attention weights in the last layer, after removing the last 8 "attention sink" tokens. The remaining attention weights are normalized to sum to 1, visualized by both the relative distance between the query and key tokens, and by the key token rank, with query positions at 512, 1024, 2048, and 4096 tokens, respectively.

averaging 20 times longer than those in WikiText. The dataset includes training, validation, and test sets with metadata, and is designed for long-range language model training. It supports open-vocabulary modeling and can be used for tasks requiring long-range reasoning.

**arXiv.** arXiv[8] is a dataset of 1.7 million arXiv articles for applications like trend analysis, paper recommender engines, category prediction, co-citation networks, knowledge graph construction and semantic search interfaces.

---

[8]https://huggingface.co/datasets/arxiv-community/arxiv_dataset

**SlimPajama.**  The SlimPajama-627B[9] dataset, hosted by Cerebras, is a cleaned and deduplicated version of the RedPajama dataset. It includes 627 billion tokens sourced from Common Crawl, C4, GitHub, and other datasets.  The dataset is designed for large-scale language model training and includes train, validation, and test splits, with detailed metadata for each text.

**LongBench.**  LongBench[10] is the first benchmark for bilingual, multi-task, and comprehensive assessment of long context understanding capabilities of large language models. It consists of various natural language processing tasks, including question answering, summarization, and text generation, with both English and Chinese language support. The dataset contains multiple subsets specifically designed to test models' abilities to handle long-range dependencies in text, making it suitable for evaluating models on tasks requiring extended context comprehension.  In the following, we describe the datasets we used from LongBench.

**Qasper.**  QASPER[11] is a dataset for question answering on scientific research papers. It consists of 5,049 questions over 1,585 Natural Language Processing papers. The dataset supports a range of question types, including factual, comparison, and clarification queries, making it suitable for training and evaluating models that need to comprehend scientific texts.

**MultiFieldQA.**  The MultiFieldQA dataset is a part of the LongBench benchmark, designed to test models' ability to answer questions based on long articles from diverse fields. These articles include sources like research papers, legal documents, government reports, and more. The dataset includes two versions: MultiFieldQA-en (in English) and MultiFieldQA-zh (in Chinese).  Questions in this dataset are manually annotated by experts, making it suitable for evaluating models on long-context question-answering tasks, where the goal is to comprehend and extract relevant information from extended texts.

**HotPotQA.**  HotPotQA[12] is a question-answering dataset with 113,000 Wikipedia-based question-answer pairs.  It emphasizes multi-hop reasoning, requiring models to extract information from multiple documents to answer a single question. The dataset also includes sentence-level supporting facts, enabling explainable reasoning, and contains comparison questions to assess the ability to compare facts across documents.  It is designed for diverse, challenging QA tasks that involve complex reasoning over long text passages.

**2WikiMultihopQA.**  2WikiMultiHopQA is a question-answering dataset designed to test multi-hop reasoning, where answering a question requires gathering information from multiple Wikipedia articles.

**GovReport.**  GovReport[13] is a large-scale collection of detailed reports from the U.S. Government Accountability Office and Congressional Research Service, each accompanied by a human-written summary, spanning a wide variety of national policy issues.

**MultiNews.**  MultiNews[14] is a large-scale dataset for multi-document summarization, containing news articles and their human-written summaries. Each summary in the dataset is generated from multiple news articles, making it ideal for tasks involving synthesizing information from diverse sources into a cohesive summary. The dataset helps evaluate the ability of models to handle multi-document summarization, a more complex form of text summarization than single-document approaches.

**TREC.**  TREC (Text REtrieval Conference)[15] is a question classification dataset used to train models for question type prediction. The dataset is valuable for evaluating few-shot question answering systems by testing their ability to classify questions into the correct type for further processing.

---

[9]https://huggingface.co/datasets/cerebras/SlimPajama-627B
[10]https://huggingface.co/datasets/THUDM/LongBench
[11]https://huggingface.co/datasets/allenai/qasper
[12]https://huggingface.co/datasets/hotpotqa/hotpot_qa
[13]https://huggingface.co/datasets/ccdv/govreport-summarization
[14]https://huggingface.co/datasets/alexfabbri/multi_news
[15]https://huggingface.co/datasets/CogComp/tre

**TriviaQA.** TriviaQA[16] is a large-scale question-answering dataset that includes over 650K question-answer pairs. The questions are sourced from trivia competitions, and the dataset contains evidence documents from Wikipedia and the web to support the answers. TriviaQA is designed to evaluate models' ability to perform reading comprehension and answer questions based on long, multi-sentence documents. It includes both unfiltered and web-filtered versions, supporting various QA tasks.

**PassageCount.** PassageCount seeks to create a more demanding situation where the model is required to utilize the full context to resolve the task. Each piece of data was generated by randomly selecting several passages from English Wikipedia, repeating each paragraph at random several times, and finally shuffling the paragraphs.

**PassageRetrieval.** The PassageRetrieval dataset in LongBench is a synthetic task designed to evaluate a model's ability to retrieve specific passages. For each entry, 30 passages are sampled, and one is summarized using GPT-3.5-Turbo. The task challenges models to identify the original passage that matches the generated summary, testing long-context understanding and passage retrieval capabilities.

**LCC.** The Microsoft LCC (Long Code Completion)[17] dataset is designed for code completion tasks and is available in multiple programming languages, including Python, Java, and C#. It is part of a series of datasets aimed at evaluating the ability of machine learning models to predict the next line of code in long programming contexts. The dataset is split into training and test sets, making it useful for training models like transformers for code generation or code completion tasks.

**RepoBench-P.** RepoBench-P (Pipeline)[18] is a part of the RepoBench dataset, which is designed to evaluate repository-level code auto-completion systems. It combines two tasks: code retrieval and code completion. First, the model retrieves the most relevant code snippet from another file (cross-file context), and then it predicts the next line of code based on that retrieved context. RepoBench-P is particularly useful for assessing the performance of models in real-world multi-file programming scenarios, where code dependencies span multiple files. The dataset is available for Python and Java.

## C  EXPERIMENT DETAILS

### C.1  REPRODUCIBILITY

**Code.** The code for the experiments is provided in the supplementary material with a well-written README file. We also provide the commands and instructions to run the code. We also provide instructions on downloading and pre-processing datasets to convert them to binary files for accelerated computation.

**Environment.** We conducted all our experiments on an Ubuntu 22.04 machine with 640GB RAM and 8 NVIDIA H100 GPUs, each equipped with 80GB of graphic memory, connected via HBM3. The code for our algorithms is written in Python (version 3.11.9). To run the code, several additional libraries are required, including PyTorch, Huggingface Transformers, Accelerate, and DeepSpeed. For detailed instructions, please refer to our README and setup.py in the code directory.

We have optimized our code and tested that the space cost of the GPU memory is less than 80 GB during SPARSE TRAINING. The execution time to run a post-training experiment is less than 16 hours on our machine.

---

[16]https://huggingface.co/datasets/mandarjoshi/trivia_qa
[17]https://huggingface.co/datasets/microsoft/LCC_python
[18]https://huggingface.co/papers/2306.03091

## C.2 IMPLEMENTATION DETAILS AND HYPERPARAMETERS

We use AdamW optimizer with warmup_min_lr, warmup_max_lr, warmup_num_steps, and total_num_steps set to "auto" in deepspeed. The default choices of hyperparameters in our code are provided in Table 8. For initializing LLaMA-2-7B, we use the default LLaMA config[19].

Table 8: Default hyperparameters for the SPARSE TRAINING

| Hyperparameter | Meaning | Value |
|---|---|---|
| batch_size | The batch size for training | 1 |
| criterion | The criterion for calculating loss | "cross_entropy" |
| learning_rate | The learning rate for optimizer | 0.00001 |
| $\beta$ | Ratio of mixed training | 1 |
| allgather_partitions | whether to use allgather | "true" |
| allgather_bucket_size | Size of allgather communication chunks | 2e8 |
| gradient_accumulation_steps | # Gradients to combine before updating weights | 1 |

## C.3 TRAINING CURVES

We record and report the training curves in Figure 8, Figure 9 and Figure 10. Figure 8 shows the perplexity while extending the context window to 8192. Due to space limitations, we only plot the first 100 steps. First, the training perplexity (loss) decreases in general and seems to be more converged as the training goes on. Second, by only ten training steps, SPARSE TRAINING is able to efficiently decrease the training perplexity from over 100 to nearly 11. Figure 9 shows the training perplexity while extending the context window from 8192 (8K) to 10240 (10K). Similar properties can also be identified. Figure 10 shows the training perplexity (loss) of the whole post-training progress. A scatter of extending window size $K$ and training perplexity $p$ means that, the training perplexity at the *last step* among the 1000 steps that extend the context window to $K$ is $p$. Although the training is continuous, the model must adapt to the new context window size each time it is extended. As a result, perplexity does not decrease monotonically. However, the overall training perplexity gradually decreases over the course of post-training, without the spikes in perplexity seen in the vanilla LLaMA-2-7B model as context window length increases, demonstrating the effectiveness of our SPARSE TRAINING.

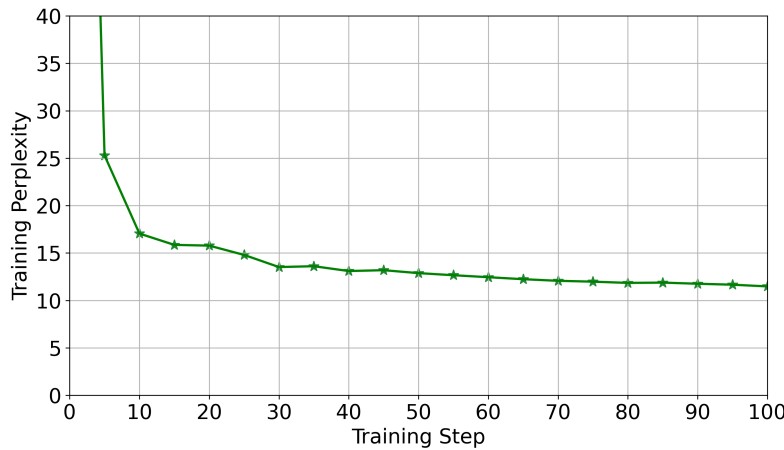

Figure 8: Training perplexity ($e^{loss}$) when extending LLaMA-2-7B to 8192 context window using SPARSE TRAINING. The perplexity converges quickly from $\sim 130$ to $\sim 11$.

---

[19]https://huggingface.co/docs/transformers/main/model_doc/llama2

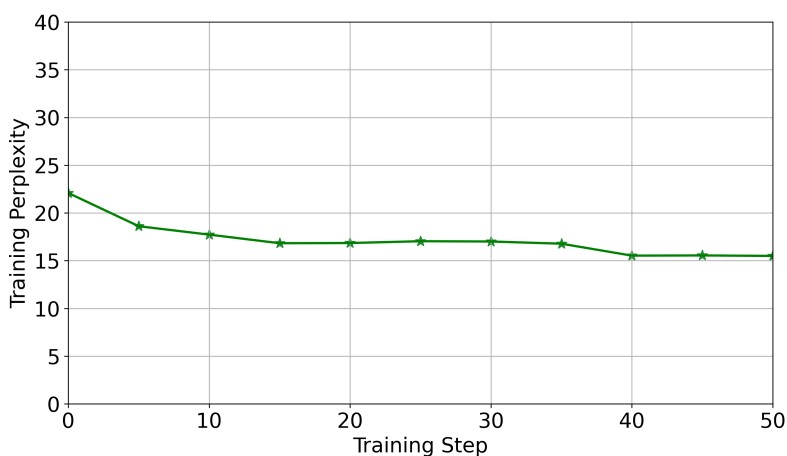

Figure 9: Training perplexity ($e^{loss}$) when extending LLaMA-2-7B from 8192 to 10240 context window using SPARSE TRAINING. The perplexity decreases from $\sim 22$ to $\sim 15$ in only 50 steps.

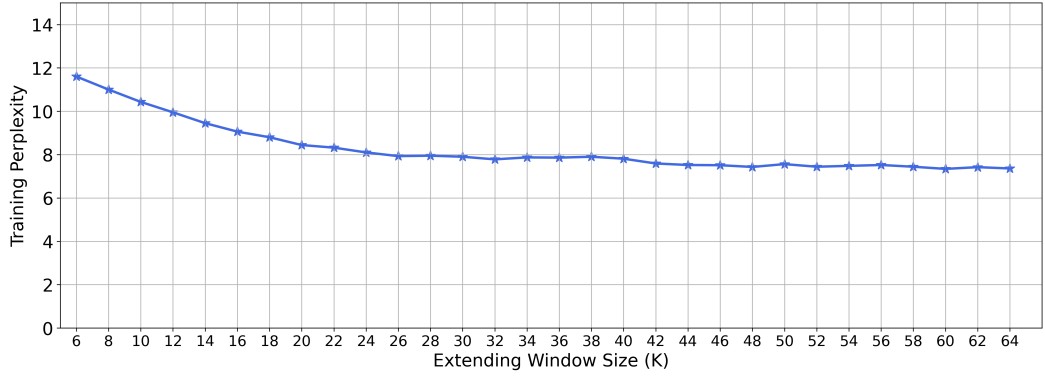

Figure 10: Converged training perplexity ($e^{loss}$) when extending LLaMA-2-7B context window using SPARSE TRAINING. While the perplexity of vanilla LLaMA-2-7B would explode over window size, the converged perplexity of each extending keeps decreasing with the context window.

## C.4 Perplexity on Longbench Datasets

Table 9: Perplexity (↓) of LLaMA-2-7B on LongBench datasets. The performance of LLaMA-2-7B after SPARSE TRAINING is stable and improves with longer contexts.

| Model | Context Length | Single-Doc QA | | Multi-Doc QA | | Summarization | |
|---|---|---|---|---|---|---|---|
| | | Qasper | MultiFieldQA | HotPotQA | WikiMQA | GovReport | MultiNews |
| Vanilla | 8K | 186.86 | 114.16 | 107.63 | 122.47 | 90.22 | 116.24 |
| | 16K | 1430.46 | 1014.01 | 948.14 | 991.11 | 949.16 | 1045.72 |
| | 32K | 3274.92 | 3207.46 | 3082.40 | 3300.21 | 5355.33 | 2983.43 |
| | 64K | 8048.33 | 4306.92 | 6096.41 | 6253.19 | 15710.44 | 5334.57 |
| Sparse Training | 8K | 13.23 | 11.47 | 9.87 | 10.48 | 9.46 | 9.71 |
| | 16K | 11.66 | 9.37 | 8.14 | 8.43 | 8.67 | 9.26 |
| | 32K | 9.56 | 6.93 | 6.6 | 6.63 | 8.13 | 7.93 |
| | 64K | 7.98 | 5.57 | 5.72 | 6.19 | 7.7 | 8.9 |

| Model | Context Length | Few-shot Learning | | Synthetic Task | | Code Completion | |
|---|---|---|---|---|---|---|---|
| | | TREC | TriviaQA | PassageCount | PassageRetrieval | LCC | RepoBench-P |
| Vanilla | 8K | 102.56 | 142.23 | 115.59 | 153.44 | 69.78 | 84.12 |
| | 16K | 1055.62 | 1121.32 | 839.71 | 1146.01 | 1051.47 | 1164.55 |
| | 32K | 3786.72 | 2895.37 | 2613.10 | 2977.54 | 2973.48 | 3050.69 |
| | 64K | 7541.48 | 5640.73 | 6844.86 | 7631.39 | 4820.02 | 5122.14 |
| Sparse Training | 8K | 7.87 | 11.15 | 10.51 | 16.86 | 4.98 | 5.13 |
| | 16K | 6.90 | 9.28 | 9.12 | 13.56 | 4.73 | 5.26 |
| | 32K | 5.43 | 7.24 | 8.39 | 11.24 | 2.56 | 2.48 |
| | 64K | 5.30 | 7.31 | 7.60 | 10.04 | 2.28 | 2.26 |

## C.5 Full results of Ablation Study

Table 10: PPL (↓) of SPARSE TRAINING without mixed training (regularization) on LongBench.

| Model | Context Length | Single-Doc QA | | Multi-Doc QA | | Summarization | |
|---|---|---|---|---|---|---|---|
| | | Qasper | MultiFieldQA | HotPotQA | WikiMQA | GovReport | MultiNews |
| Sparse Training w/o Mixed Training | 16K | 1332.19 | 1134.13 | 1806.22 | 1933.51 | 751.38 | 1005.19 |
| | 32K | 1471.31 | 1269.43 | 2870.13 | 4162.76 | 5829.82 | 5470.50 |
| | 64K | 310.02 | 368.17 | 433.99 | 418.16 | 608.48 | 1135.98 |

| Model | Context Length | Few-shot Learning | | Synthetic Task | | Code Completion | |
|---|---|---|---|---|---|---|---|
| | | TREC | TriviaQA | PassageCount | PassageRetrieval | LCC | RepoBench-P |
| Sparse Training w/o Mixed Training | 16K | 130.55 | 1868.78 | 1859.02 | 2073.48 | 1247.91 | 1058.37 |
| | 32K | 121.05 | 1295.00 | 3274.54 | 3270.87 | 1507.53 | 1498.12 |
| | 64K | 654.35 | 570.09 | 457.65 | 451.15 | 302.14 | 292.75 |

## D Limitations

SPARSE TRAINING still has some limitations, which can be summarized as follows: 1) SPARSE TRAINING focuses solely on reducing the quadratic complexity of the Transformer network during training, while it still suffers from quadratic complexity during the inference stage. Therefore, we may need to combine other inference tricks to address this inherent issue of Transformer; 2) Specifically, SPARSE TRAINING enables models to learn more semantic information from unseen positional information, rather than context information from long sequences. However, we think that this problem can be alleviated if we can determine which memory part is more important to the target part, and leave this part as future work.

