# OpenReview forum: "Sparse Training: Do All Tokens Matter for Long Sequence Generalization?"
_ICLR.cc/2025/Conference — ICLR 2025 Conference Withdrawn Submission_

### Official Review · Reviewer_x17W · 2024-10-29

**Soundness:** 2
**Presentation:** 3
**Contribution:** 2
**Rating:** 5
**Confidence:** 3

**Summary:**

This work addresses the quadratic computational cost of transformers during training, which is especially pronounced with long inputs. The authors introduce a sparse training technique to reduce training costs, motivated by the observation that there is typically minimal attention between distant tokens (Fig. 2). In their training technique, they partition inputs into a memory part (tokens distant from recent tokens) and target part (recent tokens). Tokens are then sampled from the memory part using a decaying sampling probability, with each sampled token’s positional encoding left unchanged (i.e., each token retains the positional encoding it’d have in the full input sequence). These sampled memory tokens are concatenated with the untouched target tokens, and the model performs standard next-token prediction on the target part only (c.f., Eq. 2). The authors show that their sparse training technique improves a transformer’s (i.e., LLaMA-2-7B) long-context capabilities by fine-tuning its key and query weights (Sec. 4).

**Strengths:**

* **S1**: Moving the sparsity from the architecture to the input is a very interesting approach and represents a plug-and-play change in the training of LLMs. Importantly, it is architecture-invariant, as it is merely a training technique that operates as a form of sparsity input augmentation and can be easily integrate into the standard LLM training pipeline.

* **S2**: The authors show that their simple yet effective training technique substantially improves long-range performance (Sec. 4.1), roughly maintains the performance of the pre-trained LLM (Sec. 4.2), and is also computationally cheap (Sec. 4.3), as it maintains the same computational and memory cost, as the window size is unchanged.

* **S3**: Code is provided, which helps in reproducibility.

**Weaknesses:**

* **W1**: The experiments in Sec. 4 are only conducted on LLaMA-2-7B. It would be beneficial to include other models, such as the other LLMs shown in Fig. 2 (i.e., GPT-2 & Mistral-7B). Given that the authors use these LLMs as motivation for their proposed sparse training technique but did not fine-tune them raises concerns about the method’s effectiveness beyond LLaMA-2-7B. However, I am confident that their main findings will also hold for these LLMs. Thus, I encourage the authors to include experimental results for them as well.

* **W2**: The sparse training technique is not compared to relevant baselines. For example, it would be important to compare it to methods that extend positional encodings, like YaRN. Without such comparisons, it is difficult to thoroughly assess the effectiveness of the proposed training technique. Comparisons to the vanilla LLMs (before fine-tuning) are not sufficient for a comprehensive assessment.

* **W3**: While the quadratic cost is mitigated during training, it is still present during inference. Thus, end users that use LLMs only in inference mode don’t directly benefit from the authors’ approach.

* **W4**: LLMs appear to be inherently limited by the maximal number of tokens that they were trained on. The proposed training technique extends that but probably (see **Q5**) won’t help, as we go beyond the maximum input sequence lengths seen during the fine-tuning. This contrasts with methods like YaRN, which extends LLMs beyond their maximal input sequences seen during training/fine-tuning. As a result, we must know a priori the maximum input sequence lengths that are used during inference, which may not always be foreseeable at training/fine-tuning time.

* **W5** (related to **W2** & **W4**): I believe that the reason for the training technique's effectiveness is that it simply exposes positional encodings from long contexts. Specifically, previously out-of-distribution (OOD) positional encodings are made in-distribution (ID) through the fine-tuning. Under this interpretation, the proposed sparse training technique isn’t particularly novel from a higher level, as it is well-known across many domains (e.g., domain generalization) that making OOD samples ID is a “simple trick” for better generalization.

* **W6**: The lemmas (Lemma 3.1 & 3.2) seem unnecessary and straightforward. They don’t add significant value to the current work.

**Questions:**

* **Q1**: Do the authors keep the positional encodings unchanged in the vanilla models in their experiments? Why not interpolate rotary positional encodings (RoPE) as done in YaRN? It would be meaningful to include such a comparison.

* **Q2**: Have the authors also attempted sparse training for LLMs from scratch (e.g, using a smaller pretraining dataset to reduce computational demands)?

* **Q3**: Why do the authors sample individual tokens rather than small chunks (i.e., a small set of sequential tokens)?

* **Q4**: How does the fine-tuned LLM perform on “needle in a haystack” problems or passkey retrieval tasks?

* **Q5**: How does the fine-tuned LLM perform for sequence lengths >>64k tokens? It would be interesting to compare the model’s current performance in such longer contexts (without further fine-tuning).

---

### Official Review · Reviewer_NNCT · 2024-11-01

**Soundness:** 2
**Presentation:** 2
**Contribution:** 2
**Rating:** 3
**Confidence:** 4

**Summary:**

This paper propose Sparse Training, a training technique aimed to efficiently enhance the long-context capabilities of large language models. During fine-tuning, it randomly samples tokens from the long-context to form a synthetic training sequence. This operation shortens the context window and improves training efficiency.

**Strengths:**

1. This paper focus on the important topic of extending the context window of LLMs.
2. The author clearly states the method with both and figure

**Weaknesses:**

1. The experiment setting is quite limited. The author only conduct experiments to show perplexity and next-word prediction accuracy. On one hand, recent studies have shown that perplexity is not a reliable metric for long-context modeling [1, 2]. On the other hand, the next-word prediction accuracy is not a widely accepted metric for the evaluation of language models. I strongly recommend the authors to provide results on long-context benchmarks. I note that the author have given the next-word prediction accuracy on the datasets in LongBench, why not following their pipeline and present the task scores? Such experimental results will greatly influence my assessment of the quality of this paper; otherwise, I cannot be convinced of the effectiveness of this method.
2. Insufficient discussion to related work [3,4], which I find them similar to this work. (Also see question 2)
3. It is well-known that Llama-2-7B cannot do language modeling beyond 4k tokens, so it is an unfair baseline for long-context modeling. Maybe the author can treat [3,4] as the baselines.
4. The author claims that sparse training saves a lot of training time. However, I notice that sparse training requires 30k steps of fine-tuning, which takes 16h in total using 8 H100 GPUs. From my experience, other long-context fine-tuning methods like YaRN only takes about 20h to fine-tune using 8 A100 GPUs, since it takes much less training steps (0.4k steps) than sparse training. In this case, the training efficiency advantage brought by sparse training becomes less apparent. Besides, sparse training utilizes PEFT to accelerate training, but the author does not conduct ablation study on its importance.
5. I find it rather off-putting that the authors present two trivial conclusions, lacking mathematical formulation, as lemmas; moreover, they do not serve any subsequent theorems.

[1] Hu, Yutong, et al. "Can Perplexity Reflect Large Language Model's Ability in Long Text Understanding?." arxiv.org/abs/2405.06105

[2] Hsieh, Cheng-Ping, et al. "RULER: What's the Real Context Size of Your Long-Context Language Models?." arxiv.org/abs/2404.06654

[3] Ruoss, Anian, et al. "Randomized positional encodings boost length generalization of transformers." arxiv.org/pdf/2305.16843

[4] Zhu, Dawei, et al. "Pose: Efficient context window extension of llms via positional skip-wise training." arxiv.org/pdf/2309.10400

**Questions:**

1. I wonder why there are white white strips (both vertical and diagonal) in Figure 2&4, since you have average over thousands of samples? If the strips are caused by attention on certain informative tokens, they should be diminished over the massive sample. Besides, I don't understand the meaning of the white diagonal strips.
2. I am still not convinced of the importance of the so-called "sparsity" in your design. In my understanding, sparsity means that a token will only attend to very few tokens in the long context, and intuitively these tokens are very important for long-context modeling. However, how can you guarantee that you can sample out those important tokens in your training? If cannot, I fail to see the significance of the "sparsity" in your method. In this way, I think your method is similar to the ones I mentioned in weakness 2, since you claimed that your method is superior because it achieves sparsity.

---

### Official Review · Reviewer_4KUs · 2024-11-02

**Soundness:** 2
**Presentation:** 2
**Contribution:** 1
**Rating:** 3
**Confidence:** 5

**Summary:**

The paper addresses the long-context problem for transformer-based language models (LMs). Its goal is to reduce the quadratic complexity of attention (only during training), thereby enabling the extension of a pre-trained LM’s context length beyond its original limit. The core approach is to split the given context into a memory and target part, where memory includes subsampled distant tokens, and the target is the remainder of the context where regular next-token prediction is applied. Sparse sampling of distant tokens is justified by observing the sparsity pattern in full attention. Experimental results demonstrate the method’s effectiveness in adapting a pre-trained LM for long contexts.

**Strengths:**

- The sparse training method is intuitive and well-explained.
- Experiments demonstrate the effectiveness of the proposed method in extending a model (e.g., Llama 2) with a 4k context length to support longer contexts (up to 64k).
- Experimental results support the claim that training time complexity remains constant as context length increases.

**Weaknesses:**

A major concern with this paper is the lack of baseline comparisons. All experimental results are compared only to the “vanilla” approach, which involves taking a pretrained model with a 4k context length and evaluating it on tasks requiring a longer context. As highlighted in the introduction and related work sections, several existing methods address this problem. Some interpolation methods, like Yarn and NTK, can be applied even without any training. For example, increasing LLaMA’s Rotary frequency by a factor of 10 or more is achievable with a single line of code change and would offer a stronger baseline than the reported “vanilla” approach. Similarly, window-attention could serve as a baseline without additional training compute overhead—by continuing training LLaMA 2 (trained with a 4k context length) using a window attention of the same width (4k). Other existing methods also prune the context with heuristics rather than random sampling, as proposed in this work. For example see [1] and baselines there. Overall, assessing the effectiveness of the proposed method is challenging without comparing it to existing baselines.

- Another major issue with the proposed method is that it applies only at training time, leaving the inference-time complexity still quadratic, which significantly limits its alignment with the initial motivation. How would the model perform if similar sampling were also applied during inference? In the abstract, the paper is motivated by long-context generation (rather than long-context comprehension). However, the proposed method primarily addresses long-context comprehension, with no results provided on long-context generation.

- In Lines 186-188, it is mentioned that a challenge of window-attention methods is their lack of access to distant tokens (beyond the window size), while Figure 3’s analysis suggests those tokens are important. However, this view is not entirely clear. Models **trained** autoregressively with window attention can learn mechanisms to retain a long history, even with limited window attention. This concept also motivates the NoPE method.

- Section 4.2 aims to demonstrate that the original pre-trained LLM performance is maintained through the proposed sparse training. However, the results provided are limited to perplexity. This claim should be supported by standard language model evaluations, as done in the original LLaMA 2 paper, using benchmarks like MMLU, HellaSwag, ARC-Easy/Hard, and Winograd, among others.

- Section 4.4: The ablation study on mixed training effectiveness lacks clarity.
  - Mixed training is introduced to prevent forgetting on regular-length sequences (e.g., 4k for LLaMA 2). However, the ablation results only cover 16k to 64k sequences, omitting shorter lengths like 8k, 4k, and below. This ablation should include results for regular-length sequences (both perplexity and task-specific evaluations, such as MMLU, ARC, HellaSwag, etc.) comparing models trained with and without mixed training.
  - Why does omitting mixed training significantly affect long-context performance? The impact should ideally be limited to regular sequences (<4k context).
  - The reported perplexities in the ablation are only for the case without mixed training, with the comparison data placed in the Appendix. Both sets of results should be presented in the same table for clarity.


[1] Anagnostidis, Sotiris, et al. "Dynamic context pruning for efficient and interpretable autoregressive transformers." Advances in Neural Information Processing Systems 36 (2024).

**Questions:**

- Lines 146-148: The explanation is vague. What exactly is $\alpha_{(i)}$? What information does $S_k$ convey about token $x_k$?
- Figure 4: What causes the diagonal artifacts?
- Table 1: How is accuracy defined? Is it for the next-token prediction task, and what would be the expected performance with random guessing?
- Equation (1): What role does the term  $\frac{1}{N}$ play?
- Line 138: Attention averaging is stated as being over each layer and sample. Is this averaging also applied across different heads?
- Line 144: It’s mentioned that “attention distribution always exhibits sparsity when processing long sequences.” Is this truly always the case, or could it depend on the underlying data? For example, in code-related domains, would attention remain sparse and localized?
- Table 4: Consider removing Table 4 and simply adding a new row in Table 2 instead.
- Line 429: There’s a typo.

---

### Official Review · Reviewer_YuaN · 2024-11-04

**Soundness:** 2
**Presentation:** 2
**Contribution:** 2
**Rating:** 3
**Confidence:** 4

**Summary:**

This paper focuses on efficient long-sequence training. The authors first find that tokens in the front positions contribute less to the forward propagation of the current token, as indicated by the low attention score. Therefore, the authors propose to apply a sparse sampling policy that decays with the distance from the target token, to obtain conditional tokens, named Sparse Training. The proposed method is claimed to be able to extend existing Transformer-based LLMs to capture long-range dependencies within a fixed window size during the training, which will reduce the training cost to a large extent. Experimental results on multiple datasets seem to demonstrate the effectiveness and efficiency of sparse training in long-context modeling.

**Strengths:**

* The paper studies how to effectively extend the context window of LLM, which addresses a key challenge in long-context modeling.
* The background and proposed method are presented clearly and concisely.
* The proposed method appears to be original.

**Weaknesses:**

There are several main weaknesses in the paper, including:
* Improper and inadequate evaluation. This includes: 1) The paper compares the proposed method solely to LLaMa-7B, which lacks any length extrapolation capability. To provide a fair comparison, the authors should include other length extrapolation methods, such as Yarn [1] and LongQLora [2], as baselines. 2) Limited evaluation metrics. The paper evaluates its method using LongBench, but there are issues with metric selection and interpretation. Specifically, the results suggest that the authors may be focusing on “LongBench accuracy”—the accuracy of next-word prediction—instead of using the standard LongBench score, which is more suited to evaluating question-answering performance. For instance, the paper reports a PassageCount score of 0.5 compared to the previous state-of-the-art of 0.05, which appears overly high. This limited evaluation does not sufficiently demonstrate the model's performance on long-context tasks.
* Confusing motivation. The authors claim that the tokens in the front positions contribute little to the forward propagation of the current token. However, it is known that modeling long-range dependency is crucial in long-context modeling. The proposal to uniformly sample tokens from front positions as input conditions may overlook significant information. This seems to be detrimental to long-context modeling.

Additionally, there are some minor weaknesses:
* Unnecessary lemma. Lemmas 3.1 and 3.2 do not contribute to any subsequent theorems, and no proofs are provided for these lemmas.
* Inconsistent Formatting. The font of subscripts is not uniform; for instance, $L_{sample}$ on line 90 differs from $X_{target}$ on line 226. It is advisable to standardize the formatting for clarity.

[1] YaRN: Efficient Context Window Extension of Large Language Models, arXiv:2309.00071

[2] LongQLoRA: Efficient and Effective Method to Extend Context Length of Large Language Models, arXiv:2311.04879

**Questions:**

In addition to the questions in the weakness part, I have some additional questions:
* The author claims that the proposed method is efficient. However, with using PEFT and 8 H100 GPUs, it takes 16 hours in the finetuning stage, which does not show any superiority compared to other finetuning technique like YaRN. Again, can the authors provide comparison with more baselines other than only with LLaMa 7B?
* Are there any ablation studies for the selection of the hyperparameters $\beta$ in Equation (3)?
* The authors claim that the proposed method is orthogonal to the previous length extrapolation methods. So I guess the model can benefit from joint utilization of the proposed method and the previous method. Can the authors provide experiment results for this claim?

---

### Note · Authors · 2024-11-20

I have read and agree with the venue's withdrawal policy on behalf of myself and my co-authors.